# SEQUENTIAL DIFFUSION LANGUAGE MODELS

## ABSTRACT

Diffusion language models (DLMs) have strong theoretical efficiency but are limited by fixed-length decoding and incompatibility with key-value (KV) caches. Block diffusion mitigates these issues, yet still enforces a fixed block size and requires expensive training. We introduce Next Sequence Prediction (NSP), which unifies next-token and next-block prediction, enabling the model to adaptively determine the generation length at each step. When the length is fixed to 1, NSP reduces to standard next-token prediction. Building on NSP, we propose Sequential Diffusion Language Model (SDLM), which can retrofit pre-trained autoregressive language models (ALMs) at minimal cost. Specifically, SDLM performs diffusion inference within fixed-size mask blocks, but dynamically decodes consecutive subsequences based on model confidence, thereby preserving KV-cache compatibility and improving robustness to varying uncertainty and semantics across the sequence. Experiments show that SDLM matches or surpasses strong autoregressive baselines using only 3.5M training samples, while achieving 2.1× higher throughput than Qwen-2.5. Notably, the SDLM-32B model delivers even more pronounced efficiency gains, demonstrating the strong scalability potential of our modeling paradigm. Code and models will be released.

## 1 INTRODUCTION

In recent years, diffusion models have made significant progress in computer vision, dominating various fields such as image generation (Ho et al., 2020; Rombach et al., 2022) and robot control (Chi et al., 2023; Kapelyukh et al., 2023). This successful paradigm has recently emerged as a potential solution for language modeling, *i.e.,* diffusion language models (DLMs). Compared to autoregressive language models (ALMs), DLMs generate tokens in parallel through a denoising process, showing superior theoretical efficiency. However, DLMs are also criticized for its fixed decoding length and inability to use KV cache (Radford et al., 2019).

To address these limitations, it is a natural thought to combine the benefit of DLM and ALM, similar to existing efforts like Block Diffusion (Arriola et al., 2025). Specifically, Block Diffusion reformulate the next token prediction of ALM as the next block prediction, where tokens in each block are decoded in a diffusion manner. In this case, Block Diffusion not only preserve the autoregressive property for flexible and robust prediction, while also exploiting diffusion-style parallel prediction for better efficiency.

Despite the effectiveness, Block Diffusion models still remain two practical limitations. Firstly, the block size is fixed in block diffusion models, which means that the model should predict a constant number of tokens in each step. However, the distribution of certainty and semantics varies across the entire sequence, typically requiring adjusting the suitable block size in predicting different subsequences. As shown in Figure 1(b), a fixed block size easily fails in token prediction that requires previous context information. Secondly, both the DLM and the block diffusion model require training from scratch and cannot be easily developed from a pre-trained ALM. This not only leads to significant training costs but also creates obstacles for developing larger models.

In this paper, we introduce *Next Sequence Prediction* (NSP), a general form of next token prediction and next block prediction. Specifically, NSP defines an autoregressive probability distribution for sequences of discrete random variables. As shown in Figure 1(c), NSP predicts future sequences of variable length, where a sequence can be either one token or a block of tokens. At each step, NSP decodes the tokens in the sequence in a diffusion manner. Therefore, NSP can dynamically adjust

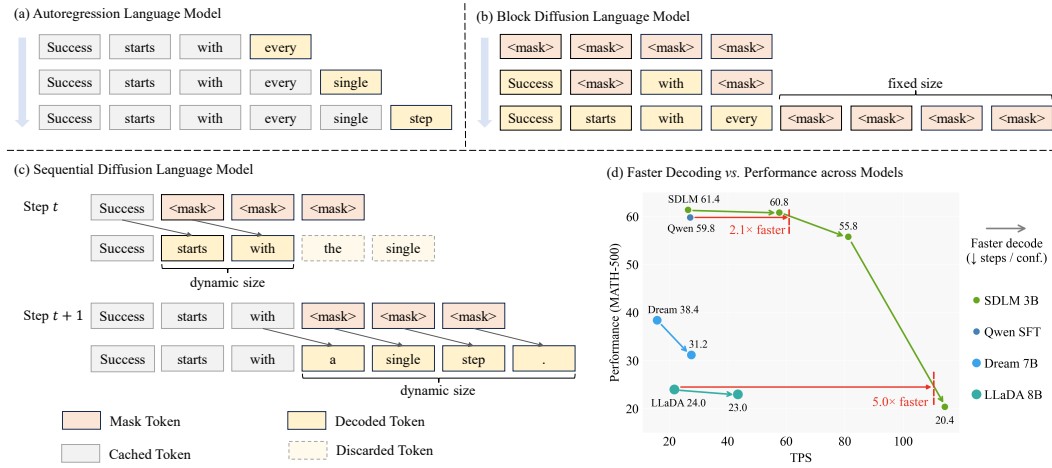

Figure 1: **Comparison of decoding paradigms.** (a) ALMs: decode one token at a time. (b) DLMs (*e.g.* Block Diffusion): decode all tokens in a fixed block before moving to the next. (c) SDLM (Ours): dynamically predicts a contiguous subsequence within a fixed block. (d) Performance *vs.* Speed: MATH-500 results showing trade-off between speed (TPS) and accuracy.

its decoding sequence size according to the difficulty and semantics of future sequences. When the length of the prediction sequence is always 1, NSP degenerates to next-token prediction. This property allows NSP to seamlessly adapt to existing pre-trained ALMs with cheap costs.

Based on the principle of NSP, we propose *Sequential Diffusion Language Models* (SDLMs) with innovative training and inference strategies. As shown in Figure 2, SDLMs are developed based on a pre-trained ALMs, employing a novel parallel block training approach to extend next token prediction to next sequence prediction. In parallel block training, we use a custom attention mask that makes the prefix and the current block visible to each prediction window, enabling parallel training over multiple future blocks. During inference, SDLM predicts a fixed-length block at each step and then dynamically decodes a continuous subsequence via a confidence scheme based on threshold or verification. Unlike our concurrent work, Samragh et al. (2025) employs gated LoRA with next-token and multi-token prediction losses, whereas we use NTP cross-entropy loss for full supervised fine-tuning. For sampling, we apply bidirectional attention with confidence-based decoding, without extra sampling heads.

To validate our approach, we construct different scales of LLMs and conduct extensive experiments on 13 benchmarks across general, math, knowledge and coding tasks. Experiments show that our SDLMs achieve on-par performance with existing ALMs with much faster speed, *e.g.,* 2.1× faster than Qwen-2.5-3B (Team, 2024). Compared to existing DLMs, our SDLMs demonstrate comprehensive advantages in performance, efficiency, and training costs. For example, SDLM-3B significantly outperforms DLMs like Dream-7B (Ye et al., 2025) and LLaDA-8B (Nie et al., 2025b) across multiple benchmarks, while requiring far less training compute and yielding substantially higher inference speed. More importantly, the scalability of SDLMs is validated on the larger models, *i.e.,* Qwen-2.5-32B, requiring only 3.5M training samples. In summary, our contributions are three-folds:

- We introduce *Next Sequence Prediction* (NSP) as a general form of next token prediction and next block prediction. NSP not only combines the advantage of autoregressive models and diffusion models, while providing a practical way to relax the fixed block-size constraint in prior block diffusion models.

- Based on NSP, we deploy *Sequential Diffusion Language Models* (SDLMs) through a novel parallel block training method. SDLMs employ a customized attention mask where each block is visible to its prefix and itself, enabling parallel training and dynamic variable-length sequence generation via threshold- or verification-based selection.

- Extensive experiments not only demonstrate the effectiveness and efficiency against existing ALMs and DLMs, but also confirm its scalability on large-scale models. In particular, with only 3.5M training samples, our SDLMs achieves comparable performance and nearly $2\times$ speedup against Qwen-2.5-32B-SFT.

## 2 RELATED WORK

### 2.1 AUTOREGRESSIVE LLMS AND MULTI-TOKEN PREDICTION

Autoregressive large language models (ALMs), such as GPT (Radford et al., 2018; OpenAI, 2022; 2024), LLaMA series (Touvron et al., 2023a;b; Grattafiori et al., 2024; Chiang et al., 2023), Qwen series (Bai et al., 2023; Yang et al., 2024; Team, 2024; Yang et al., 2025) and other advanced LLMs (DeepMind, 2025; Team et al., 2025; xAI, 2025; Anthropic, 2023; Liu et al., 2024), generate text in a token-by-token manner and have demonstrated strong performance across a wide range of language tasks, including question answering, code generation, mathematical problem solving and dialogue systems. However, this strictly sequential decoding process limits generation speed. To mitigate this, KV Cache has been introduced to store previously computed attention keys and values, avoiding redundant computations and significantly improving inference efficiency.

To address the limitations of serial decoding, multi-token prediction (MTP) (Cai et al., 2024; Gloeckle et al., 2024; Liu et al., 2024) enables the model predict multiple future tokens in parallel via multiple output heads. These parallel predictions can be used with speculative decoding (Xia et al., 2022; Stern et al., 2018) to validate multiple candidates and greatly reduce forward steps. For example, DeepSeek-V3 shows up to $3\times$ faster inference with MTP with speculative decoding.

### 2.2 DIFFUSION LANGUAGE MODELS

Recent diffusion models have shown increasing potential in language tasks. Masked discrete diffusion models (MDMs) (Zheng et al., 2023; Gong et al., 2024; Ou et al., 2024; Nie et al., 2024) have achieved perplexity comparable to ALMs. LLaDA (Nie et al., 2025a) further scales MDMs to 8B parameters, matching state-of-the-art ALMs. Dream (Ye et al., 2025) adopts shifted prediction and autoregressive initializes, effectively reducing training costs while also delivering strong performance. Block Diffusion (Arriola et al., 2025) introduces block-level generation for variable-length decoding with KV cache reuse. Gemini Diffusion (Google DeepMind, 2025) and Seed Diffusion (Song et al., 2025) further improve speed while narrowing the gap with ALMs.

Although recent acceleration technologies such as dKV-Cache (Ma et al., 2025), Fast-dLLM (Wu et al., 2025), and dLLM-Cache (Liu et al., 2025) attempt to use approximate KV caching mechanisms to accelerate DLM inference, these methods still suffer from substantial computational overhead caused by padding the sequence to the maximum sequence length for each forward computation.

## 3 METHODS

### 3.1 PRELIMINARY AND NOTATION

In autoregressive large language models (ALMs) (OpenAI, 2024; Grattafiori et al., 2024; Yang et al., 2025; DeepMind, 2025), text generation is typically modeled as a conditional probability chain, referred to as the next-token prediction paradigm. Given a sequence of input tokens $\{x^1, ..., x^L\}$, the objective is to minimize the cross-entropy loss:

$$\mathcal{L}_{\text{ALM}}(x; \theta) = -\mathbb{E}_x \left[ \sum_{i=1}^{L} \log P_\theta(x^i \mid x^{<i}) \right], \tag{1}$$

where the model $P_\theta(\cdot \mid x^{<i})$ aims to maximizes the conditional probability of the current word by leveraging the preceding context $x^{<i} = x^0, \ldots, x^{i-1}$.

In contrast, diffusion language models (DLMs) (Ou et al., 2024; Nie et al., 2025a; Ye et al., 2025) generate outputs by progressively denoising from a fully noisy state in parallel. Block Diffusion (Arriola et al., 2025) are a specialized DLM variant that constrains the diffusion operation to proceed

sequentially in blocks. At each time step $t$, the model receives a noisy block $X_t^i = x_t^{iD:(i+1)D}$ and predicts all masked tokens (denoted as $[m]$) within a block of length $D$, formally defined as:

$$\mathcal{L}_{\text{BD}}(X;\theta) = -\sum_{i=1}^{L/D} \mathbb{E}_{t\sim[0,1]}\mathbb{E}_q \frac{\alpha_t'}{1-\alpha_t} \log P_\theta(X^i|X^{<i}, X_t^i), \tag{2}$$

where $X$ denotes the ground-truth, $q$ is the forward masking process that gradually corrupts to-kens, $\alpha_t \in [0,1]$ is the probability of keeping (not masking) a token at time $t$ so that the masking probability is $1 - \alpha_t$, and $\alpha_t'$ is the instantaneous rate of change of $\alpha_t$ in continuous time.

## 3.2 SEQUENTIAL DIFFUSION LANGUAGE MODELS

In DLMs, the entire sequence is predicted in parallel based on confidence scores. This can result in premature and inaccurate predictions for later tokens, imposing greater demands on the model's robustness. But predictions for tokens at lower position indices generally benefit from more reliable contextual information and introduce less bias (Wang et al., 2024). Meanwhile, the distribution of certainty and semantics varies across the entire sequence. To this end, we introduce the *Next Sequence Prediction* (NSP) paradigm, which aims to dynamically adjust the size of the decoding sequence at each step based on the difficulty and semantics of the future sequence.

Based on the above understanding, we propose the *Sequential Diffusion Language Models* (SDLM) to reduce error accumulation in diffusion-based generation and improve parallel prediction effi-ciency. As shown in Figure 1(c), the model adopts bidirectional attention similar to Block Diffu-sion to understand the semantic information in the future fixed-length noise block $X_T^i$. Differently, SDLM masks all tokens in the prediction block (masking probability = 1) and is trained by mini-mizing the cross entropy of all masked tokens. The training objective is formalized as:

$$\mathcal{L}(X;\theta) = -\mathbb{E}_{X,X_T} \left[ \frac{1}{D} \sum_i \log P_\theta \left( X^i \mid x^{<(i-1)}, X_T^i \right) \right],$$
$$X^i = x^{i:(i+D)}, \ X_T^i = [x^{i-1}, \underbrace{[m], ..., [m]}_{D-1}], \tag{3}$$

where $i$ denotes a random index within the target sequence, since dynamic length inference makes the decoding start position non-fixed. To better unify next token prediction and block prediction, we continue to employ standard AR's one-position shift between input IDs and labels.

During inference, we introduce *Longest Prefix Decoding*, which uses low-order position priors, to decode the next sequence based on model's confidence. Specifically, at each step, the model perceives history $x^{<(i-1)}$ and produces fixed-length future logits $Z^i = [z_i^1, \ldots, z_i^D] \in \mathbb{R}^{D\times|\mathcal{V}|}$ over vocabulary $\mathcal{V}$, ultimately decoding only the first $\gamma_\tau(Z^i)$ tokens. In the next step, predictions are repeated starting from the previous step's end position. The formalization is as follows:

$$\hat{X}^i = \text{Decode}\left(Z^i, \gamma_\tau(Z^i)\right) \tag{4}$$

where $\gamma_\tau(Z^i)$ determines the adaptive sequence length to be decoded (with $1 \le \gamma_\tau(Z^i) \le D$), and $\text{Decode}(\cdot)$ denotes extracting the next $\gamma_\tau(Z^i)$ contiguous tokens from $Z^i$, which we denote as $\hat{X}^i$. The maximum sequence length function $\gamma_\tau(\cdot)$ is detailed in Section 3.4. This adaptive length mechanism can effectively balance generation efficiency and quality based on text's semantic richness and uncertainty.

## 3.3 TRAINING

As noted in Section 3.2, when the block size is 1 our model reduces to the autoregressive paradigm, allowing reuse of pretrained ALM weights and cut training costs. From the perspective of instruction fine-tuning, we define the input as $S = [X; Y]$, where $X$ is the prefix and $Y$ the response.

**Training.** During training, we partition $Y$ into blocks at random positions to train the model's prediction capabilities at different starting positions. As shown in the Equation 3, for a starting posi-tion $i$, we construct a noise block $Y_T^i = [y^{i-1}, [m], \ldots, [m]]$ and predict the next fixed-length block

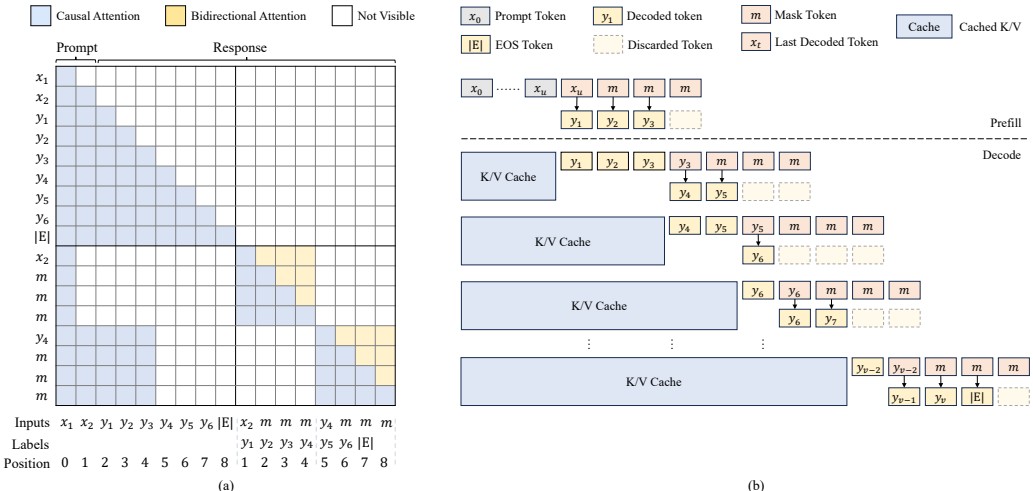

Figure 2: **Structured attention mask for parallel block training and sampling.** (a) Reordered input yields a mask with causal prefix (top-left), visible cross-block prefix (bottom-left), and intra-block bidirectional attention (bottom-right). (b) Confidence-based next sequence prediction with KV reuse. A block of $D$ tokens is predicted with $D-1$ masks. The longest high-confidence subsequence is selected as dynamic output. Cached KV states enable efficient decoding.

$Y^i = y^{i:(i+D)}$ (simplified as $Y^i = [y_i^1, \ldots, y_i^D]$) by shifting. A bidirectional attention mechanism is used within the block for feature information, which serves as the basis for decoding dynamic-length sequences. For historical information, we maintain casual attention as ALMs. Therefore, for a single noise block $Y_T^i$, we can construct a custom attention mask $A \in \{0,1\}^{(i+D) \times (i+D)}$:

$$A_{uv} = 1_{v \leq u} \oplus 1_{u \geq i \cap v \geq i} \tag{5}$$

This enforces strict causality for $u < i$ and full mutual attention for $u, v \geq i$.

**Parallel Training.** To enable efficient parallel training, we construct the sequence by interleaving noise blocks and target blocks as:

$$S_T = \text{concat}(X, \underbrace{\mathbb{I}_1 \cdot Y_T^1, \ Y^1}_{\text{Block 1}}, \ \ldots, \ \underbrace{\mathbb{I}_i \cdot Y_T^i, \ Y^i}_{\text{Block } i}, \ \ldots) \tag{6}$$

where $\mathbb{I}_i \in \{0, 1\}$ is a random indicator variable that controls whether a noise block $Y_T^i$ is inserted at the current starting position $i$ to predict the ground-truth block $Y^i$. Each noise block $Y_T^i$ attends only within itself, while $Y^i$ is visible as prefix to later blocks but not vice versa, ensuring causality through attention constraints and positional encodings.

Since transformers rely on positional encodings, by rearranging $S$, the attention mask forms three parts as shown in Figure 2: (1) causal attention (top-left), (2) visible prefixes for each block (bottom-left), and (3) bidirectional attention within blocks (bottom-right). To improve training efficiency, we can concatenate any number of noise blocks after the target sequence within max sequence length. The sparse attention structure allows flex attention (Dong et al., 2024) to accelerate training.

## 3.4 INFERENCE

As described in Equation 4, we introduce the Longest Prefix Decoding method for dynamic length decoding based on low-order position priors. We primarily rely on the model's confidence in its inferences as the basis to refine the length function $\gamma(\cdot)$, and design two types of decoding strategies:

**Greedy Decoding.** We implement $\gamma_\tau$ through a confidence-based stopping rule that identifies the longest prefix satisfying:

$$\gamma_\tau(Z^i) = \max\left(\left\{j \in \{1, 2, \ldots, D\} \mid \prod_{k=1}^{j} p(z_i^k) \geq \tau\right\} \cup \{1\}\right) \quad (7)$$

where $p(z_i^k)$ quantifies confidence at position $k$ (where $z_i^k \in \mathbb{R}^{|\mathcal{V}|}$ is the position-$k$ logit vector), and $\tau$ is a predefined threshold. This approach greedily decodes at most $j$ tokens ($j \geq 1$) whose cumulative product of confidence scores is greater than $\tau$. We explore two distinct confidence functions:

(1) Logit Value Confidence. This metric uses the softmax probability of the decoded token $v$ at position $k$, to capture the model's per-token confidence in its top prediction:

$$p_{\text{logit}}(z_i^k) = \text{softmax}(z_i^k)_v \quad (8)$$

(2) Entropy-Normalized Confidence. While $p_{\text{logit}}$ provides a pointwise confidence signal, it overlooks distributional ambiguity. Inspired by Wang et al. (2025) that higher predictive entropy correlates with forking behavior during generation, we employ an entropy-based confidence score:

$$p_{\text{entropy}}(z_i^k) = 1 - \frac{H_p}{\log|\mathcal{V}|}, \quad \text{where} \quad H_p = -\sum_{n}^{|\mathcal{V}|} p_n \log p_n \quad (9)$$

Here, $p_n$ is the softmax probability of the $n$-th word. Then, the entropy $H_p$ is normalized by $\log|\mathcal{V}|$. Lower entropy indicates higher confidence, while higher entropy reflects more uncertainty.

**Self-Speculative Decoding.** Following the speculative decoding (Stern et al., 2018), we decode multiple tokens in parallel and verify their correctness through self-consistency checks. In each step, the model produces $D$ speculative tokens $\hat{Y}^i = [\hat{y}_i^1, \ldots, \hat{y}_i^D]$ (where $\hat{y}_i^k$ denote the $k$-th decoded token of block $i$) in an initial forward pass. To validate them, $D$ verification inputs are constructed by progressively extending prefixes of the sampled tokens, appending mask $[m]$ at the first unverified position and padding to form a batch. A second forward pass then yields corresponding predictions $\widetilde{Y}^i = [\widetilde{y}_i^1, \ldots, \widetilde{y}_i^D]$. The decoding sequence length is determined by the consistency-driven function:

$$\gamma_{\text{verify}}(Z^i) = \max\left(\left\{j \in \{1, 2, \ldots, D\} \mid \hat{y}_i^j = \widetilde{y}_i^j\right\} \cup \{1\}\right) \quad (10)$$

Compared to confidence-based truncation via $\gamma_\tau$, which relies on local heuristics, self-speculative decoding performs explicit consistency checks for self-verification without external models, offering greater reliability at the cost of an additional forward pass.

## 4 EXPERIMENTS

### 4.1 SETTING

To ensure a fair comparison, we fine-tune the Qwen-2.5 base model (Team, 2024) with all open-source instruction datasets (3.5 million samples, 2.3 billion tokens), covering math, code, and instruct-following. We compare SDLM against same-scale ALMs (Qwen2.5-3B/32B-Instruct, fine-tuned verison of Qwen2.5-3B/32B under the same setting), and larger DLMs like Dream-7B-Instruct and LLaDA-8B-Instruct across benchmarks spanning general, mathematics, science, and coding tasks. All evaluated with OpenCompass (Contributors, 2023) under standardized settings. Details about training and evaluating can be found in Appendix A.

### 4.2 MAIN RESULTS

Table 1 shows the performance and inference efficiency of our SDLM, trained in a single epoch on only 3.5M samples. SDLM-32B attains 92.4 on GSM8K, 74.2 on MATH-500, and 78.6 on IFEval, while remaining competitive on coding tasks. The SDLM-3B performs on par with or even surpasses Qwen-2.5-3B-SFT, and significantly outperforms larger DLMs such as LLaDA-8B and Dream-7B.

Table 1: **Performance of instruct models across 8 long-form tasks.** Numbers in parentheses (#) denote the speedup ratio: average tokens per pass vs. ALMs (1 token per pass). Results marked by [†] and [¶] are from Team (2024) and Ye et al. (2025) respectively. "–" indicates unknown data.

| Model Name | | GSM8K | MATH | GPQA | HumanEval | HumanEval+ | MBPP | MBPP+ | IFEval | Avg. |
|---|---|---|---|---|---|---|---|---|---|---|
| | | | | | ALMs | | | | | |
| Qwen-2.5-3B[†] | | 86.7 | 65.9 | 30.3 | 74.4 | – | 72.7 | – | 58.2 | – |
| Qwen-2.5-3B-SFT | | 85.8 | 59.8 | 27.8 | 73.8 | 60.4 | 68.5 | 42.6 | 62.1 | 60.1 |
| Qwen-2.5-32B [†] | | 95.9 | 83.1 | 49.5 | 88.4 | – | 84.0 | – | 79.5 | – |
| Qwen-2.5-32B-SFT | | 93.2 | 74.8 | 33.8 | 82.9 | 76.2 | 82.1 | 59.0 | 76.5 | 72.3 |
| | | | | | DLMs | | | | | |
| LLaDA-8B[¶] | | 78.6 | 26.6 | 31.8 | 47.6 | – | 34.2 | – | 59.9 | – |
| Dream-7B[¶] | | 81.0 | 39.2 | 33.0 | 55.5 | – | 58.8 | – | 62.5 | – |
| SDLM-3B ($D = 4$) | $\tau = .98$ | 84.6 (2.15) | 60.8 (2.18) | 28.3 (2.26) | 67.1 (1.91) | 59.8 (1.76) | 65.4 (1.66) | 40.5 (1.78) | 57.1 (1.38) | 57.9 (1.89) |
| | $\tau = .82$ | 84.5 (2.75) | 57.8 (2.73) | 28.3 (2.66) | 66.5 (2.53) | 60.4 (2.25) | 65.0 (2.30) | 40.0 (2.29) | 55.8 (1.58) | 57.3 (2.39) |
| SDLM-32B ($D = 4$) | $\tau = .98$ | 92.4 (2.15) | 74.2 (2.35) | 36.4 (2.34) | 81.1 (2.05) | 73.8 (2.29) | 80.9 (1.56) | 58.2 (1.51) | 78.6 (1.25) | 71.9 (1.94) |
| | $\tau = .82$ | 92.3 (2.71) | 73.0 (2.88) | 36.9 (2.61) | 79.9 (2.82) | 73.2 (2.72) | 80.9 (2.17) | 57.1 (2.25) | 78.2 (1.43) | 71.4 (2.45) |

In terms of generation efficiency, SDLM generate about 2 tokens per forward pass, reducing latency to about two-thirds of comparable ALMs. Taking GSM8K as an example, SDLM-32B at $\tau = 0.98$ achieves accuracy 92.4 (*vs.* 93.2 for its same-scale SFT counterpart) while generating 2.15 tokens per step. Lowering $\tau$ to 0.82 further increases token output to 2.71 with only a 0.1 pct accuracy drop, highlighting an attractive speed-accuracy tradeoff. SDLM-3B follows a similar trend on GSM8K with minimal performance drop as $\tau$ is lowered. This trend holds across all benchmarks, where lowering $\tau$ consistently increases token generation while maintaining competitive performance. The effect and robustness of different $\tau$ values are ablated in Section 4.3.

In terms of short-answer benchmarks shown in Table 2, SDLM-32B performs within 1 ptc of its autogressive counterpart across MMLU, Winogrande, and Hellaswag, while SDLM-3B matches Qwen-2.5-3B-SFT on these benchmarks. This demonstrates that SDLM retains the semantic and reasoning abilities of the base ALMs while enabling more efficient parallel decoding, confirming that our diffusion training preserves the base model's NTP capability.

Overall, SDLM delivers "near-SFT accuracy with significant inference acceleration" at both 3B and 32B scales, proving that NSP generation can stably converge in large-model regimes and providing a solid foundation for future work with larger parameters, longer training, and wider blocks.

Table 2: **Performance of instruct models across 5 general mutiple-choice tasks.**

| Model Name | MMLU | Winogrande | Hellaswag | ARC-C | ARC-E |
|---|---|---|---|---|---|
| | | ALMs | | | |
| Qwen-2.5-3B-SFT | 67.6 | 60.8 | 75.3 | 83.1 | 91.4 |
| Qwen-2.5-32B-SFT | 83.7 | 78.0 | 92.4 | 94.2 | 99.1 |
| | | DLMs | | | |
| LLaDA-8B | 65.5 | – | 74.6 | 88.5 | – |
| Dream-7B | 67.0 | – | – | – | – |
| SDLM-3B ($D = 4$) | 66.3 | 60.2 | 74.2 | 82.7 | 92.0 |
| SDLM-32B ($D = 4$) | 82.8 | 79.2 | 92.0 | 94.9 | 98.9 |

## 4.3 TRADE-OFF BETWEEN SPEEDUP AND PERFORMANCE

Existing DLMs (Nie et al., 2025a; Ye et al., 2025) exploit parallel token generation but face a key trade-off: generating one token per step maintains quality, while producing multiple tokens often degrades it. Moreover, the reliance on fixed-length noise sequences constrains flexibility and limits practical efficiency gains over ALMs. In contrast, SDLM only concatenate a block-length masks per step, incurring minimal overhead compared to NTP inference.

Figure 3 shows the speed-performance trade-off with varying confidence threshold $\tau$ across GSM8K, MATH-500 and HumanEval+. As $\tau$ decreases, SDLM generates more tokens per step,

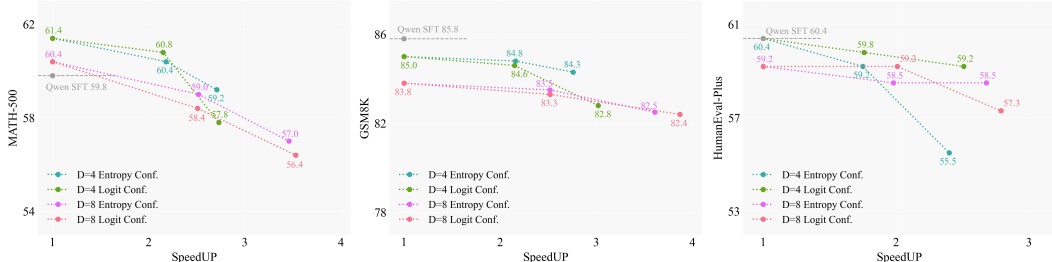

Figure 3: **Trade-off between performance and speed under different inference setting for SDLM-3B** ($D = 4$) **and SDLM-3B** ($D = 8$). Adjusting $\tau$ allows a controllable trade-off between speed and performance. SpeedUp denotes the average number of tokens output per forward pass.

Table 3: **SDLM-3B** ($D = 8$) **with larger block size and sampling with self-speculative decoding.** (a) Larger blocks yield higher throughput with only minimal performance degradation. (b) With self-speculative decoding, the average accepted tokens per step (in green) significantly exceeds greedy decoding with threshold (Conf. $\tau$).

| Model Name | | GSM8K | MATH | HumanEval+ | MBPP | MBPP+ | Avg. |
|---|---|---|---|---|---|---|---|
| Qwen-2.5-3B-SFT (AR) | | 85.8 | 59.8 | 60.4 | 68.5 | 42.6 | 63.4 |
| SDLM-3B ($D = 4$) | Conf. $\tau = .98$ | 84.6 (2.15) | 60.8 (2.18) | 59.8 (1.76) | 65.4 (1.66) | 40.5 (1.78) | 62.2 (1.91) |
| | Speculative. | 85.1 (3.62) | 61.2 (3.54) | 58.4 (3.40) | 65.8 (3.29) | 40.5 (3.23) | 62.2 (3.42) |
| SDLM-3B ($D = 8$) | Conf. $\tau = .98$ | 83.3 (2.52) | 58.4 (2.51) | 59.2 (2.01) | 64.2 (1.71) | 39.7 (2.16) | 61.0 (2.18) |
| | Speculative. | 83.6 ((5.99) | 60.2 (5.73) | 57.3 (5.18)) | 64.2 (4.84) | 39.4 (5.33) | 60.9 (5.41) |

achieving up to $3.5\times$ speed-up. On math tasks like MATH-500, accuracy remains stable ($61.4 \rightarrow 59.2$) as long as tokens per step stay under 3. Code tasks like HumanEval+ are more sensitive, with performance remaining high at around 1.7 tokens per step ($60.4 \rightarrow 59.8$).

Furthermore, we compare the effects of generation block size $D$ and confidence functions (Logit *vs.* Entropy). Results show that $D = 4$ generally yields slightly better accuracy, while the new trained model SDLM-3B ($D = 8$) enables greater acceleration due to larger parallel generation capacity. Both confidence function-based schemes maintain good performance. The threshold $\tau$ provides a flexible balance between speed and performance across various tasks and configurations.

## 4.4 ABLATION STUDY

We further explore larger $D$ and self-speculative decoding (Table 3), alternative model designs (Appendix C), and comparisons with other speed-up methods (*e.g.*, MTP; Appendix D). We retrain all variants under the same settings to assess their impact on both performance and efficiency.

**Block Size.** We investigate the impact of larger block sizes on SDLM-3B in Table 3, focusing on the new trained model SDLM-3B ($D = 8$). Compared to $D = 4$, the $D = 8$ configuration delivers substantially higher throughput with comparable model performance. Under Conf. $\tau = 0.98$, the average number of output tokens per step increases from 1.9 ($D = 4$) to 2.2 ($D = 8$), with only a 1.2-point drop in overall accuracy. increasing $D$ from 4 to 8 boosts the accepted tokens with only a small quality drop, suggesting potential for further throughput gains.

**Self-Speculative Decoding.** We further evluate self-speculative decoding in Table 3. In the *Speculative* rows, with $D = 4$ and $D = 8$, SDLM accepts an average of 3.4 and 5.4 tokens per step, corresponding to roughly 85% and 68% of the proposal budget, respectively. Model performance remains comparable across settings. However, this method incurs additional validation overhead, differing

| | |
|---|---|
| **User:** | What fraction of 2 feet is 3 inches? Express your answer as a common fraction. Please reason step by step, and put your final answer within \boxed{}. |
| **SDLM:** | To find the fraction of 2 feet that is 3 inches , we need to convert the feet to inches and then divide the  by the total number. |
| | 1 foot = 12 inches |
| | 2 feet = 2 * 12 = 24 inches |
| | Now, we can the 3 inches by the fraction of 2 4 inches: |
| | \[ \frac{3}{24} = \frac{1}{8} \] |
| | So, the answer is 1/8. |
| | \[ \boxed{\frac{1}{8}} \]<\|im_end\|>    `38 step with 113 tokens` |
| **User:** | Write a Fibonacci function in Python. |
| **SDLM:** | def fibonacci(n):    `12 step with 44 tokens` |
| |    if n == 0: |
| |      return 0 |
| |    elif n == 1: |
| |      return 1 |
| |    else: |
| |      return fibonacci(n-1) + fibonacci(n-2)<\|im_end\|>#(fib |

Figure 4: **Visualization of the sampling process.** Where each blue block indicates a subsequence generated in a single decoding step.

in experimental setup compared to the other two decoding methods. Despite this, it substantially enhances the model's responsiveness, demonstrating its potential under specific conditions.

**Case Study.** Figure 4 illustrates SDLM's flexible decoding, where the generated sequence length adapts to local context. In fluent or structured regions (*e.g.* math expressions, structured code segments, common phrases), it confidently emits longer sequences at once. While facing uncertainty or forking, it slows down with shorter outputs. This adaptive strategy balances speed with precision.

## 5 CONCLUSION

In conclusion, we propose *Next Sequence Prediction* (NSP), a unified framework bridging autoregressive and diffusion decoding. Building on NSP, we develop *Sequential Diffusion Language Models* (SDLMs) that adapt pretrained ALMs via parallel block training and dynamic decoding. SDLM matches SFT-tuned ALMs in performance while decoding faster, offering a stronger speed–performance trade-off. We hope this work inspires further exploration of unified sequence generation.

## REPRODUCIBILITY & ETHICS STATEMENT

All data are publicly available, and code is provided in an anonymous repository (https://anonymous.4open.science/r/SDLM-112358/readme.md) for full reproducibility. This work follows the ICLR Code of Ethics, upholding integrity, fairness, and respect for privacy.

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

## A    DETAILS OF TRAINING

We show the training hyperparameters in Table 4.

Table 4: **Training Hyperparameters for SDLM.**

| Parameter | SDLM-3B | SDLM-32B |
|---|---|---|
| Max sequence length | 5,632 | |
| Epochs | 1 | |
| Batch size (global) | 256 | 464 |
| Training steps | 13,699 | 7,558 |
| Learning rate | $5 \times 10^{-6}$ (constant) | |
| ZeRO stage | 1 | 3 |

The training corpus comprises with: Tulu-3-SFT-Mixture (Lambert et al., 2024), Table-GPT (Li et al., 2023), SciRIFF (Wadden et al., 2024), SmolTalk (Allal et al., 2025), OPC-SFT-Stage2 (Huang et al., 2024), and ScaleQuest-Math (Ding et al., 2024), with a combined total of 3.5 million samples ($\sim$ 2.3 billion tokens).

To comprehensively evaluate the capabilities of SDLM, we conduct evaluations across a diverse set of benchmarks encompassing:

**General Tasks.** MMLU (Hendrycks et al., 2021a)(5-shot), Winogrande (Sakaguchi et al., 2021)(0-shot), Hellaswag (Zellers et al., 2019)(10-shot), ARC-C/E (Clark et al., 2018)(0-shot), IFEval (Zhou et al., 2023)(0-shot).

**Mathematics & Science Tasks.** GSM8K (Cobbe et al., 2021) (0-shot), MATH-500 (Hendrycks et al., 2021b)(0-shot), GPQA (Rein et al., 2024) (0-shot).

**Coding Tasks.** HumanEval (Chen et al., 2021) (0-shot), Humaneval+ (Liu et al., 2023) (0-shot), MBPP (Austin et al., 2021) (3-shot), MBPP+ (Liu et al., 2023) (3-shot).

## B    PSEUDOCODE FOR SDLM

We now include pseudocode for SDLM training (Algorithm 1), confidence-based decoding (Algorithm 2), and self-speculative decoding (Algorithm 3). The corresponding input IDs, position IDs, and attention masks are shown in Figure 2 (a) (training) and Figure 5 (inference). For confidence-based decoding, the relevant mask is the bold upper-left block in Figure 5.

---

**Algorithm 1** Supervised Fine-Tuning of SDLM

**Require:** Model $\mathcal{M}$, Sequence $x$, block size $D$
**Ensure:** Parameters $\theta$
 1: **repeat**
 2:     $S_T \leftarrow$ construct input via Eq. 3, Eq. 6
 3:     $S_0 \leftarrow$ ground-truth labels
 4:     PE $\leftarrow$ block-wise position ids aligned with $S_T$
 5:     A $\leftarrow$ Fig. 2 (a)
 6:     $\hat{S}_0 = \mathcal{M}(S_T, \text{ PE, A})$
 7:     $\mathcal{L}(S_T, S_0) = \text{CE}_{\text{block}}(\hat{S}_0, S_0)$
 8:     $\theta \leftarrow \theta - \eta \nabla_\theta \mathcal{L}$
 9: **until** converged
10: **return** $\theta$

---

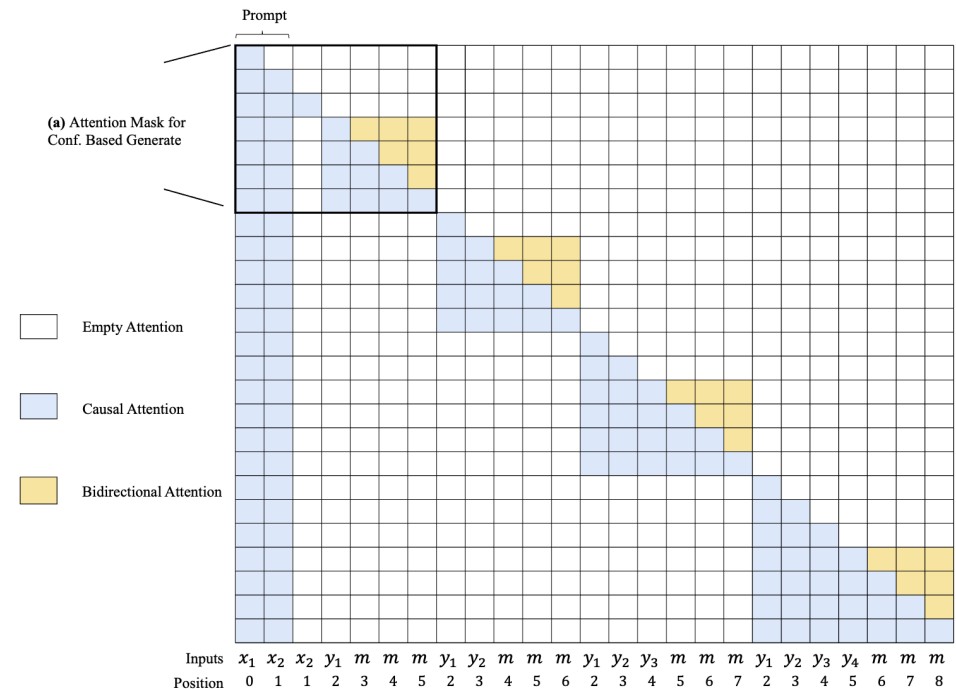

Figure 5: **Structured attention mask for sampling.** (a) The bold upper-left block shows the mask used during confidence-based generation; the full mask shows the mask used in self-speculative decoding.

---

**Algorithm 2** Conf Generate

---

**Require:** Model $\mathcal{M}$, prompt $x$, block size $D$
**Ensure:** Generated sequence $y$

1: Initialize KV-cache; $\quad y \leftarrow x; \quad L \leftarrow |x|$
2: **while** not EOS **do**
3: $\quad x_{\text{in}} \leftarrow [\, y, \ y_{-1}, \ \text{MASK}^{\times(D-1)} \,]$ $\qquad\qquad\qquad\qquad$ ▷ prepare diffusion block
4: $\quad \text{PE}_{\text{in}} \leftarrow \text{range}(0, L) \parallel [L{-}1] \parallel \text{range}(L{-}1, \ L{+}D{-}2)$ $\qquad$ ▷ revise pe for
5: $\quad (\text{logits}, \text{KV}_{\text{new}}) \leftarrow \mathcal{M}(x_{\text{in}}, \text{KV}_{\text{cache}}, \text{PE}_{\text{in}})$
6: $\quad \text{KV}_{\text{cache}} \leftarrow$ truncate $\text{KV}_{\text{new}}$ to positions $0{:}L$ $\qquad\qquad$ ▷ only update causal KV
7: $\quad \hat{d}_{1:D} \leftarrow$ last $D$-token predictions from logits $\qquad\qquad\qquad$ ▷ candidate block
8: $\quad k \leftarrow \max\{j \in [1, D] \mid \text{confidence}(\hat{d}_{1:j}) \geq \tau\}$ $\qquad$ ▷ longest prefix accepted
9: $\quad y \leftarrow y \parallel \hat{d}_{1:k}; \quad L \leftarrow L + k$ $\qquad\qquad$ ▷ append accepted tokens
10: $\quad$ **if** $\hat{d}_{1:k}$ contains EOS **then**
11: $\qquad$ **break**
12: $\quad$ **end if**
13: **end while**
14: **return** $y$

---

## C    MORE ABLATION STUDY OF MODEL DESIGN

**No Shift Prediction.** To verify the effectiveness of shift prediction, we employ a method similar to LLaDA to directly predict the original tokens at the mask location. As shown in Figure 6, under the same training cost, this method leads to a noticeable decline in model performance, with Hu­manEval+ scores dropping by approximately 14 points. After log analysis, we find that the model has more repeated outputs. This indicates that the shift prediction method exploits the strong ability of ALMs to predict the first token and provides a stable starting point for diffusion decoding.

---

**Algorithm 3** Self-Speculative Decoding Generate

---

**Require:** Draft/Verify model $\mathcal{M}$, prompt $x$, block size $D$
**Ensure:** Generated sequence $y$

1: Initialize KV-cache;   $y \leftarrow x$;   $L \leftarrow |x|$
2: pad_len $= D(D-1) + \frac{D(D+1)}{2}$

3:                                                                                   ▷ **Prefill: obtain initial draft block $y_{1:D}^{\text{draft}}$**
4: $x_{\text{in}} \leftarrow [\, y,\ y_{-1},\ \text{MASK}^{\times(D-1)}\,]$
5: $\text{PE}_{\text{in}} \leftarrow \text{range}(0, L) \,\|\, [L-1] \,\|\, \text{range}(L-1, L+D-2)$
6: $(\text{logits}, \text{KV}) \leftarrow \mathcal{M}(x_{\text{in}}, \text{PE}_{\text{in}}, \text{KV})$
7: $\text{KV} \leftarrow$ truncate to positions $0{:}L$                                       ▷ keep only causal cache
8: $y^{\text{draft}} \leftarrow$ last-$D$ predictions from logits

9:                                                                                   ▷ **Decode: draft & verify loop**
10: **while** not EOS **do**
11:    $(\,\text{pad}(y^{\text{draft}}),\ \text{pad}(\text{PE})\,) \leftarrow \text{PadInputs}(y^{\text{draft}}, L)$       ▷ block-pyramid verify input, Fig. 5
12:    $x_{\text{in}} \leftarrow [\, y,\ y_{-1},\ \text{pad}(y^{\text{draft}})\,]$
13:    $\text{PE}_{\text{in}} \leftarrow \text{range}(0, L) \,\|\, [L-1] \,\|\, \text{pad}(\text{PE})$
14:    $(\text{logits}, \text{KV}_{\text{new}}) \leftarrow \mathcal{M}(x_{\text{in}}, \text{PE}_{\text{in}}, \text{KV})$   ▷ joint forward: compute verify + new draft blocks
15:    $\text{KV} \leftarrow$ truncate $\text{KV}_{\text{new}}$ to positions $0{:}L$                              ▷ only update causal part
16:    $y^{\text{verify}} \leftarrow$ last-pad_len predictions from logits
17:    $(\, y^{(0)},\ y^{(1)}\,) \leftarrow \text{reshape}(y^{\text{draft}}, y^{\text{verify}})$   ▷ draft block candidates & verify block candidates
18:    $(\, y^{\text{new-draft}},\ k\,) \leftarrow \text{AcceptCount}(y^{(0)}, y^{(1)})$                      ▷ determine accepted tokens
19:    $y \leftarrow y \,\|\, y_{1:k}^{\text{draft}}$;    $L \leftarrow L + k$;
20:    **if** $y_{1:k}^{\text{draft}}$ contains EOS **then**
21:       **break**
22:    **end if**
23:    $y^{\text{draft}} \leftarrow y^{\text{new-draft}}$                                          ▷ update draft block candidates
24: **end while**
25: **return** $y$

---

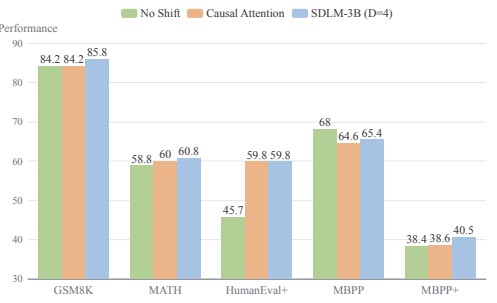 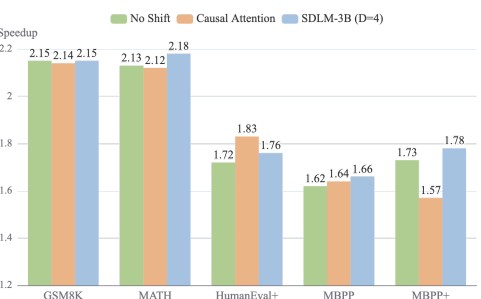

Figure 6: **Ablation on attention mask type and prediction shift strategy.** We conduct the following ablation experiments: (1) No shift: predicting $x_t$ instead of $x_{t+1}$; (2) Leisure precautions: using a causal mask instead. The left image shows its model performance, while the right image shows the acceleration ratio.

**Causal Attention.**    As shown in Figure 6, we replace bidirectional attention inside each block with a causal (unidirectional) masking. With a block size $D = 4$, the two variants obtain almost identical scores on some benchmarks and exhibit comparable training difficulty. However, the average number of tokens generated per step decreases from $1.88$ to $1.82$, indicating that bidirectional attention enlarges the local receptive field during decoding and improves parallel generation efficiency.

Table 5: **SDLM with larger block size and sampling with self-speculative decoding.** SC.: Performance; SuP (SpeedUp×): average accepted tokens per step; TPS: actual throughput, denotes as Equation 11. In addidion, BD denotes the length of draft tokens (num of heads in MTPs and decoding window size of SDLM).

| Model | | GSM8K SC. | SuP | TPS | MATH SC. | SuP | TPS | HumanEval+ SC. | SuP | TPS | MBPP SC. | SuP | TPS | Avg. SC. | SuP | TPS |
|---|---|---|---|---|---|---|---|---|---|---|---|---|---|---|---|---|
| ALMs | | | | | | | | | | | | | | | | |
| Vicuna-7B-v1.5[1] | Vanilla | 11.8 | 1.00 | 40.00 | 2.0 | 1.00 | 40.26 | 16.5 | 1.00 | 39.01 | 38.1 | 1.00 | 40.21 | – | 1.00 | 39.87 |
| Qwen2.5-3B-SFT | Vanilla | 86.0 | 1.00 | 30.95 | 60.8 | 1.00 | 31.22 | 61.0 | 1.00 | 31.32 | 68.5 | 1.00 | 31.00 | 69.1 | 1.00 | 31.12 |
| Qwen2.5-3B-SFT | vLLM | 85.6 | 1.00 | 138.15 | 62.2 | 1.00 | 138.29 | 59.2 | 1.00 | 137.05 | 70.0 | 1.00 | 132.46 | 69.3 | 1.00 | 136.49 |
| MTPs | | | | | | | | | | | | | | | | |
| Medusa-V1.0-7B[2] | BD=6 | 10.4 | 3.64 | 119.15 | 2.0 | 3.82 | 128.40 | 16.5 | 3.90 | 127.74 | 26.1 | 4.25 | 103.40 | – | 3.90 | 119.67 |
| Qwen2.5-3B-Eagle3 | BD=16 | 77.41 | 5.78 | 142.29 | 57.60 | 6.13 | 182.31 | 59.15 | 5.71 | 146.37 | 67.70 | 5.24 | 139.39 | 65.45 | 5.72 | 152.59 |
| Qwen2.5-3B-Eagle3 | BD=32 | 78.51 | 8.01 | 256.97 | 59.60 | 8.65 | 190.39 | 58.54 | 7.73 | 161.99 | 68.48 | 7.18 | 157.33 | 66.28 | 7.89 | 191.67 |
| DLMs | | | | | | | | | | | | | | | | |
| SDLM-3B | BD=4 | 85.0 | 3.62 | 98.52 | 60.4 | 3.57 | 98.58 | 59.2 | 3.44 | 93.49 | 65.4 | 3.35 | 86.11 | 67.5 | 3.50 | 94.18 |
| ($D = 4$) | BD=8 | 85.5 | 5.34 | 143.30 | 60.0 | 5.12 | 137.49 | 59.8 | 4.88 | 129.43 | 65.4 | 4.51 | 111.31 | 67.7 | 4.96 | 130.38 |
| SDLM-3B | BD=8 | 83.6 | 5.99 | 161.00 | 59.6 | 5.73 | 155.53 | 57.9 | 5.37 | 141.47 | 64.2 | 5.09 | 125.27 | 66.3 | 5.55 | 145.82 |
| ($D = 8$) | BD=16 | 84.2 | 7.30 | 176.44 | 60.2 | 6.82 | 166.83 | 59.2 | 6.33 | 152.69 | 65.4 | 5.68 | 126.39 | 67.3 | 6.53 | 155.59 |

# D  COMPARISON WITH MULTI-TOKEN PREDICTION

SDLM can be viewed through the lens of multi-token prediction (MTP) as well. Both SDLM and MTP parallelize autoregressive generation by predicting multiple tokens in a single forward pass. For a prediction horizon of $D$ tokens, MTP use $D$ separate output heads, with the $i$-th head predicting the token at position $m + i$. Similarly, SDLM uses $D$ positions in the input sequence: the last token (at position $m$) and $D - 1$ mask tokens. The prediction at the last token position corresponds to the token at $t + 1$ (equivalent to MTP's first head), and the prediction at the $j$-th mask token ($1 \leq j \leq D - 1$) corresponds to the token at $m + 1 + j$ (equivalent to MTP's $(j + 1)$-th head).

However, SDLM introduces two advantages. First, the predictions are generated within a local bidirectional attention window, enabling joint context utilization across the predicted tokens. This contrasts with MTP's isolated head (Cai et al., 2024; Gloeckle et al., 2024) or left-to-right attention (Liu et al., 2024). Second, extending the prediction horizon requires no architectural modification: appending additional mask tokens suffices, while MTP necessitates adding new output heads.

To ensure a fair comparison with MTP style methods with speculative decoding, we evaluate the following models:

- Autoregressive: both vanilla HuggingFace `transformers.generate()` (PyTorch backend with pre-allocated KV-cache) and the vLLM Kwon et al. (2023) acceleration framework;

- Medusa Cai et al. (2024): A method accelerating generation by employing a few additional decoding heads; evaluated using Medusa-V1.0-7B and its base model, Vicuna-7B-v1.5;

- Eagle-3 Li et al. (2025): A SoTA multi-token prediction with speculative decoding; evaluated using trained Qwen2.5-3B-Eagle3 with our training data;

- SDLM (Ours): Our method of self-speculative decoding with enhanced KV-cache support.

We compute TPS with actual generated tokens and wall-clock inference time as follows:

$$\text{TPS} \approx \frac{\sum_{\text{num\_samples}} \text{actual\_generate\_tokens}}{\text{wall-clock inference time}} \tag{11}$$

As shown in Table 5, we first compare SDLM with the SFT baseline model trained under the same setting, SDLM exhibits only a small drop in performance across the four benchmarks, but already achieves about a $3\times$ end-to-end speedup over vanilla decoding, when both the training and inference

---

[1] https://huggingface.co/lmsys/vicuna-7b-v1.5

[2] https://huggingface.co/FasterDecoding/medusa-v1.0-vicuna-7b-v1.5

windows size are set to 4. Furthermore, for SDLM-3B ($D = 8$), when we increase the inference window size to 16, we observe an even longer effective decoding length without noticeable performance degradation, and the actual speedup reaches about $5\times$ compared with vanilla decoding, while remaining highly comparable to vLLM across the four evaluated tasks.

For comparison with MTP approaches, Medusa achieves nearly a $3\times$ speedup over its base model, and the state-of-the-art speculative decoding model Eagle-3 also demonstrates close to a $5\times$ speedup (see their Table 1 on GSM8K for a fair comparison). In our evaluation, when using a block window of 16, SDLM achieves acceleration comparable to Eagel-3 (TPS 155 *vs.* 152), while exhibiting slightly lower speedup compared with Eagle-3 using a larger block size of 32.

These results indicate that SDLM can achieve excellent acceleration when combined with speculative decoding. Moreover, SDLM naturally generalizes to larger speculative windows that are never seen during training, and the bidirectional attention design may further enhance modeling capacity. We believe that with more carefully optimized training objectives and attention kernels, SDLM can become even faster and more accurate in the future.

All results in Table 5 are evaluated using the standard OpenCompass settings, and the wall-clock inference time is measured by simply placing a timer around the `generate` call.

## E   USE OF LARGE LANGUAGE MODELS

We used large language models (LLMs) solely for assisting with language polishing and minor writing support. All research ideas and analyses were conceived and developed by the authors.

