# OpenReview forum: "Sequential Diffusion Language Models"
_ICLR.cc/2026/Conference — Submitted to ICLR 2026_

### Official Review · Reviewer_TpGK · 2025-11-01

**Soundness:** 3
**Presentation:** 3
**Contribution:** 3
**Rating:** 4
**Confidence:** 4

**Summary:**

This paper introduces sequential diffusion language models that (1) reformulate the training objective from block diffusion that all the inputs in the block are now masked tokens, and (2) enable parallel training through a specially designed attention mask. During inference, the authors employ greedy decoding and self-speculative decoding to generate and sample tokens in parallel. The results show that it can have strong performance compared to its base model and faster inference speed.

**Strengths:**

1. The paper proposes two novel language models that enable parallel decoding while achieving superior performance compared to previous diffusion language models.
2. The proposed approach is methodologically sound, and the resulting models demonstrate strong empirical performance.

**Weaknesses:**

1. I was wondering if this model can still be called a diffusion model anymore. The only thing left in the current DLM that is related to the term `diffusion` is that they randomly mask different tokens at different timesteps. However, according to the training objective proposed in this paper in equation 3, it omits all the timestep-related design and adds several mask tokens together. It has no relationship with the theory of diffusion models. Can the author clarify why this model can still be called a diffusion model?
2. Important comparisons and ablations are missing. The most straightforward baseline to compare with is the block diffusion. The training objective and baseline idea are derived from that paper, but no comparison is shown. What benefits can we get from this new training objective? What would be the effect if we removed the timestep-dependent random masking?
3. Missing comparison with multi-token prediction and speculative baselines. This paper can be very similar to the idea of extending an AR model to predict multi-tokens (Medusa series) and then using/not using speculative decoding (the self-speculative decoding part). Do you have any comparisons with those baselines?

**Questions:**

1. How much extra training overhead would be introduced by applying this special attention mask? Since the token length would be longer than the original sequence.
2. Why would the performance of Math be higher than LLaDA/Dream by a large margin? I guess it's not the benefits from the new training recipe, but maybe some different tricks in the instruction tuning stage

---

> ### Author Response · Authors · 2025-11-21
>
> We sincerely thank the reviewer for valuable comments and provide our responses and clarifications below.
>
> **Is SDLM A Diffuiosn Language Model?**
>
> In continuous DMs, the process is defined with explicit timesteps and Gaussian noise. In contrast, discrete diffusion for language is typically implemented through masked diffusion: masking tokens and training a denoiser to recover them. Recent work shows that the diffusion NELBO in this setting is equivalent to a weighted sum of MLM cross-entropy losses, without requiring an explicit timestep variable (MD4 [1]: “a weighted integral of cross-entropy losses”; MDLM [2]: “a mixture of classical MLM losses”).
>
> SDLM directly follows this masked-diffusion formulation:
> * Forward: mask a block of tokens;
> * Reverse: predict the original tokens from mask tokens.
>
> Thus, although SDLM uses a single full-mask corruption per block and omits explicit timesteps, its objective is exactly the standard masked diffusion objective in cross-entropy form. SDLM is therefore a blockwise, single-step masked diffusion language model, fully aligned with recent discrete diffusion theory.
>
> [1] Shi, Jiaxin, et al. "Simplified and generalized masked diffusion for discrete data." NIPS 2024.
>
> [2] Sahoo, Subham, et al. "Simple and effective masked diffusion language models." NIPS 2024.
>
> **Comparisons with BD-3LM**
>
> As you noted, our method is inspired by BD-3LM. However, our goal is to convert existing AR models into DLMs with minimal changes, so we further simplify both the training objective and the decoding strategy, including shifted prediction and dynamic-length decoding.
>
> Reply to Question 1: We also experimented with the exact BD-3LM objective in early explore. Although those experiments used slightly different datasets, the results remain representative: under the same training budget, our approach better preserves the original autoregressive capabilities and achieves overall stronger performance (see Table below). Notably, on MBPP, BD3LM showed instruction-following issues, for example producing “[BBEGIN]” instead of “[BEGIN]”—suggesting potential biases or misalignment in certain cases.
>
> > Comparison of SDLM with BD-3LM. The mask block length is 4, and the training data and number of epochs are kept consistent.
>
> |Model|MATH|GSM8k|GPQA|HumanEval|MBPP|
> |--|--|--|--|--|--|
> |BD-3LM-D4|49.8 |69.8|15.7 |59.2|38.2|
> |SDLM-D4  |49.4 |71.3|20.2 |59.8|47.2|
>
> Reply to Question 2: We removed the timestep-dependent random masking for two reasons.
> 1. Decoding coherence: Utterances vary in semantic density, and forcing multi-step decoding at a fixed length can introduce truncation or redundant reasoning. Our dynamic-length strategy keeps only the longest prefix from a single decoding step, which better preserves coherence.
> 2. Training efficiency: SDLM trains on a single fully masked state, allowing all tokens in the block to be supervised simultaneously. This provides dense, consistent signals, simplifies the objective, and leads to faster convergence in practice.
>
> **Comparisons with MTP**
>
> We appreciate the suggestion and have added comparisons with MTP and speculative decoding baselines. SDLM delivers substantial speedup, with only a slight performance drop compared to the SFT AR model. It benefits from bidirectional attention and larger block sizes, enabling strong end-to-end efficiency. Even against SoTA speculative decoding such as Eagle-3, SDLM remains competitive, and we expect further improvements with optimized diffusion-style inference kernels.
>
> For completeness, we provide the full experimental details, settings, and additional results in the new updated Appendix D of the revision.
>
> **Training overhead**
>
> Although the special attention mask increases effective sequence length, SDLM requires only single SFT stage (no additional pretraining or multi-stage diffusion training). The total cost remains modest:
> * AR SFT: $\approx$ 660 GPU hours;
> * SDLM: $\approx$ 2000 GPU hours.
>
> This is far smaller than prior  DLMs such as Dream (over 24,000 GPU hours, also initialized from Qwen). Thus, SDLM adds some overhead but remains extremely lightweight relative to existing diffusion-based models.
>
> **Performance of Math**
>
> The performance gap primarily stems from:
> 1. Training data. Our SFT uses ScaleQuest-Math-1M, improving both SFT and SDLM reasoning ability. SDLM’s masked training does not degrade autoregressive reasoning.
> 2. Inference limitations of DLMs (e.g., LLaDA). LLaDA use fixed decoding lengths (e.g., 512 tokens), causing many math solutions to be truncated. SDLM generates until 'EOS' within each block, avoiding truncation.
>
> These factors explain SDLM’s stronger math performance without relying on special instruction-tuning tricks.

---

### Official Review · Reviewer_en7T · 2025-11-02

**Soundness:** 2
**Presentation:** 2
**Contribution:** 2
**Rating:** 6
**Confidence:** 3

**Summary:**

This paper proposes sequential diffusion models. SDLMs perform inference by predicting a block of $D$ tokens ahead in parallel, all at once, from masked tokens. This gives more FLOPs per prediction than eg multi-token prediction. Predictions are then truncated based on sequence confidence (Longest Prefix Decoding). A couple options are explored for confidence functions: product of probabilities, product of entropy-related quantities, and a self-consistency check drawn from Blockwise Parallel Decoding. Training is the same as Block Diffusion, relying on fixed block sizes.

Results demonstrate that SDLMs offer speedups over the original autoregressive model. No comparisons are made to speculative decoding or multi-token prediction methods.

**Strengths:**

The method leads to speedups over sequential decoding, but so will any speculative decoding method with a decent acceptance rate. The writing was clear.

**Weaknesses:**

The main weakness is a lack of comparison to spec decoding methods. I assume this weakness has been anticipated by the authors. Since speculative decoding has been optimized by the community, I would not expect SDLMs to be faster on wall-clock time. A reasonable win would be to show an improved accept length per prediction. Faster wall-clock time as well would of course be great. Eagle-3 claims to have speedups of around 3.5-4.4x and accept lengths around 5-6 tokens on Llama 8b. The results in Table 3 are not far off in terms of accept lengths, but are not an apples-to-apples comparison.

One main benefit of diffusion models is any-order inference, e.g. infilling. It's possible that if pure spec-decoding cannot be beat in the sequential setting, infilling may be the main benefit of blending diffusion models + speculative inference.

**Questions:**

The main discussion point is comparison to speculative decoding methods. In the weaknesses section, I suggested comparing to Eagle 3.

---

> ### Author Response · Authors · 2025-11-21
>
> We sincerely thank the reviewer for the very insightful and constructive comments.
>
> In our newly added experiments in Appendix D, we find that the actual inference speed of SDLM is already comparable to SoTA speculative decoding methods, e.g. Eagel-3. As shown in Appendix D Table 5, with decoding window size = 16, SDLM with self-speculative decoding achieves close to 5× end-to-end speedup (TPS 155 vs. 31 of vanilla AR), while incurring only very minor performance degradation. Eagle-3 also demonstrates close to a 5× speedup (see their Table 1 on GSM8K for a fair comparison). We believe that the bidirectional attention in SDLM is particularly well suited for denoising relatively long draft outputs, and our training observations are also consistent with Eagle-3: *"with more training steps, the average accepted length per decoding step increases"*, suggesting that both larger block size and longer training can further improve speed and accuracy in the future.
>
> In contrast, Eagle-3 is fundamentally a **draft/verify speculative decoder**, whose performance ceiling is determined by its underlying target autoregressive model. If the goal is strictly lossless acceleration without changing the original model distribution, then Eagle-3 is indeed a very strong and often preferable choice.
>
> However, SDLM and more broadly DLMs represent a different modeling direction. While DLMs are still relatively early in their development, their architecture naturally supports parallel decoding and bidirectional context modeling, which leaves more room for future gains: on the one hand, more efficient attention kernels and system-level optimizations can further improve inference speed; on the other hand, bidirectional attention and any-order generation provide stronger modeling flexibility and enable richer application scenarios. From this perspective, we believe the SDLM still offers unique and promising advantages in terms of long-term capability and task diversity, which we view as highly worthwhile to explore.

---

### Official Review · Reviewer_pJp5 · 2025-11-09

**Soundness:** 2
**Presentation:** 3
**Contribution:** 3
**Rating:** 4
**Confidence:** 4

**Summary:**

This work proposes Sequential Diffusion Language Models (SDLM), a framework that aims to unify autoregressive decoding and diffusion decoding, through a next sequence prediction (NSP) objective. SDLM/NSP adopts block-wise token generation (e.g., generating 4 token at a time). It further uses a confidence-based or consistency-based length function to identify a usable, longest prefix of the generation -- making the generation block-wise but variable length as well. SDLM is continued trained from regular autoregressive LLMs (e.g., Qwen-2.5). The experiments compare SDLM with autoregressive and diffusion LMs, showing inference speedup with reasonable performance as well.

**Strengths:**

This paper addresses an important problem in block-wise token generation in LLMs, making the usually fixed block size variable-length. The proposed SDLM can be continued to be trained from regular LLMs. The experiments show reasonable speed-ups for inference. The writing is clear and well-organized.

**Weaknesses:**

(1) Relevance to diffusion models. Though this work frames SDLM as a diffusion model, it doesn't seem to have a "diffusion" procedure (e.g., with a noising procedure, interpolating between a clean and noise distribution, etc.). Instead, each prediction block is always fully masked with a masking probability of 1. This makes the connection to diffusion language models questionable. The prediction length in each block is also much shorter (D=4).

(2) Baselines in the multi-token prediction domain. This work compares with regular language models (Qwen) and also two diffusion language models (LLaDA, Dream). However, this work should have a stronger connection with the multi-token prediction works. Direct comparisons with Medusa [1] and work like [2] should be most relevant.

(3) The performance drop. The performance drop of SDLM at the same model size of regular LMs is non-trivial. Table 1 shows that when D=4, in average there's a 2.2 to 7.3 points performance drop. Though it offers 1.89 to 2.39x speed-ups, this gap needs more justification. Also, does average tokens per pass directly translate to the same amount of acceleration in reality? (e.g., need to factor in advanced inference frameworks like vLLM, speculative decoding, etc?)

[1] Cai et al., 2024. Medusa: Simple LLM Inference Acceleration Framework with Multiple Decoding Heads.

[2] Gloeckle et al., 2024. Better & Faster Large Language Models via Multi-token Prediction.

**Questions:**

Does the method work for D > > 8? It could be hard for general tasks I assume, but would it work for tasks that require large chunks of copying?

---

> ### Author Response · Authors · 2025-11-21
>
> We sincerely thank the reviewer for valuable comments and provide our responses and clarifications below.
>
> **Relevance to diffusion models**
>
> In continuous DMs, the process is defined with explicit timesteps and Gaussian noise. In contrast, discrete diffusion for language is typically implemented through masked diffusion: masking tokens and training a denoiser to recover them  from mask token. Recent work shows that the diffusion NELBO in this setting is equivalent to a weighted sum of MLM cross-entropy losses, without requiring an explicit timestep variable (MD4 [1]: "a weighted integral of cross-entropy losses"; MDLM [2]: "a mixture of classical MLM losses").
>
> SDLM directly follows this masked-diffusion formulation:
> * Forward: mask a block of tokens;
> * Reverse: predict the original tokens from mask tokens.
>
> Thus, although SDLM uses a single full-mask corruption per block and omits explicit timesteps, its objective is exactly the standard masked diffusion objective in cross-entropy form. SDLM is therefore a blockwise, single-step masked diffusion language model, fully aligned with recent discrete diffusion theory.
>
> Regarding the concern about the block size D=4, we clarify the following:
> 1. The validity of a  DLM does not depend on a specific D. SDLM can in principle use larger blocks; we used D=4/8 mainly due to compute limits. Increasing D from 4 to 8 already preserves accuracy while improving speed, suggesting the formulation is stable.
> 2. As shown in our new Appendix D, using a larger block at inference than in training also works well: performance remains stable while speed improves, indicating good generalization to larger effective block sizes.
>
> Thus, the experiments with 4 and 8 are sufficient to show SDLM is a valid diffusion LM and that its block size can be scaled further.
>
> [1] Shi, Jiaxin, et al. "Simplified and generalized masked diffusion for discrete data." NIPS 2024.
>
> [2] Sahoo, Subham, et al. "Simple and effective masked diffusion language models." NIPS 2024.
>
> **Effective in the MTP domain**
>
> We now include direct comparisons with MTP and speculative decoding baselines. SDLM delivers substantial speedup, with only a slight performance drop compared to the SFT AR model. Its bidirectional attention and larger block sizes further improve efficiency over multi-token prediction. SDLM also remains competitive with the SoTA Eagel-3, and we expect future DLM infrastructure improvements to further boost speed and capability.
>
> Additional experimental details and results are provided in the new updated Appendix D.
>
> **Performance Drop and acceleration in reality**
>
> 1. Performance: We acknowledge a *calculation error* and apologize for the oversight. The correct average is **57.3**, so the performance drop is smaller and more moderate than previously stated.
> 2. SpeedUP vs. wall clock accelerate: As discussed in Appendix D, SpeedUP closely matches actual TPS in our self-speculative decoding setup: although each step handles longer sequences, the >6× reduction in model forward passes dominates latency. This alignment mirrors prior speculative decoding results (e.g., Eagle-3).
> 3. Comparison with vLLM: Our new Appendix D also provides a dedicated TPS of vLLM, SDLM achieves a peak TPS of 155, which is notably higher than vLLM’s TPS of 136 on average.
>
> **Is SDLM work for Larger $D$**
>
> Due to compute limits, we trained SDLM-3B with 4 and 8. As shown in Table 6, increasing D from $4\to8$ improves decoding speed with only a small accuracy drop on those benchmarks.
>
> More importantly, at inference time we decouple D from the decoding block size. Using larger blocks than those seen in training (e.g., $4\to8$, $8\to16$)) yields significantly higher speedups while maintaining performance, thanks to self-speculative decoding, which always verifies each block at position 1. For example, D=4 trained models achieve nearly 5× SpeedUP with inference block size 8, and D=8 trained models reach >6.5× SpeedUP (and >5× TPS).
>
> These trends hold across math and code tasks, suggesting that larger effective block sizes generalize well beyond the training D. Exploring much larger D is promising future work.
>
> On more open-ended benchmarks (IFEval, MTBench, AlpacaEval), acceleration is still substantial (3.6×–4.4×), though smaller than the 6× gains on code and math tasks, likely due to training-data distribution. For clarity, we report results on widely used benchmarks as qwen2.5.

---

### Official Review · Reviewer_Thez · 2025-11-11

**Soundness:** 2
**Presentation:** 1
**Contribution:** 2
**Rating:** 2
**Confidence:** 3

**Summary:**

The authors propose to cheaply finetune autoregressive LMs for diffusion decoding of contiguous token blocks. They extend standard block diffusion inference for variable-length block sizes, which adapts to confidence. The authors show 5x speedup over prior dLLM LLaDA with improved benchmark performance and 2x speedup over Qwen, scaling their models up to 32B parameters.

**Strengths:**

- **Technical novelty**: The authors overcome the limitations in existing block diffusion models by proposing adaptive block sizes at inference, while introducing entropy-normalized confidence and self-speculative decoding for dLLMs
- **Strong experimental results**: The authors show 5x speedup over LLaDA and 2x speedup over Qwen with limited SFT examples, scaling up to 32B parameters. Further, the benchmark performance improves substantially over prior dLLMs LLaDA and Dream. The released models will positively impact the research community
- **Thorough benchmarking and ablations**: The authors provide thorough analysis across benchmarks, showing comparable performance to AR SFT with better throughput. The authors also carefully assess the importance of block size, confidence metric, logit shifting, attention mechanism, and speculative decoding and show that their proposed recipe improves downstream performance

**Weaknesses:**

- **Unclear whether proposed Next Sequence Prediction modeling objective corresponds to a valid NELBO**
- **Ambiguous training algorithm**: It is unclear how the sequence is split into blocks at random positions (line 213), or why noised blocks are randomly inserted in the training input sequence (line 250). Training pseudocode is not provided.
- **Contiguous block decoding prevents long-range token interactions**: The proposed decoding strategy is restricted to generate contiguous blocks, preventing long-range token interactions and non-causal token conditioning, which are advantages of diffusion.
- **Missing ablation on the importance of contiguous block decoding**: It is unclear whether contiguous block decoding is necessary, which is a core contribution over [1]
- **Missing experimental comparison to multi-token prediction**: The proposed decoding strategy generates the longest contiguous block of tokens possible in each decoding step, which is very similar to multi-token prediction (MTP) approaches [2,3] as noted in Appendix B. Given their close similarities, the authors should show experimental comparison to MTP methods to better motivate their approach
- **Overstated contribution**: The authors claim that standard block diffusion training [1] is not scalable (lines 100-101), while the authors propose to overcome this by initializing from an AR checkpoint. However, the proposed training algorithm (Section 3.3) has comparable efficiency to [1]. Further, [1] could easily be directly extended to initialize from an AR checkpoint
- **The proposed parallel training is very similar to the vectorized training in [1]**, which is not cited or compared against in the text (lines 245-258). [1] also proposes parallel training to predict all blocks using FlexAttention kernels
- **Presentation/clarity**: Various parts of the paper feel rushed with numerous grammatical errors/missing definitions
  - line 96, 108: "with only 3.5M training data"
  - line 40: "benefit from the diffuse nature in efficiency"
  - line 194: "shifted-prediction objective" undefined in the section
  - line 200: $\hat{X}$ is undefined
  - line 169: $\alpha_t$ undefined and the forward noise process $q$ defined informally
  - line 214: confusing notation: $Y_T^i$ corresponds to a block of $D$ tokens, but the superscript refers to indexing at position $i$?

[1] Arriola et al. Interpolating between autoregressive and diffusion language models. ICLR 2025

[2] Cai et al. Medusa: Simple llm inference acceleration framework with multiple decoding heads. ICML 2024

[3] Gloeckle et al. Better & Faster Large Language Models via Multi-token Prediction. arXiv 2024

**Questions:**

- Can the authors provide pseudocode for the training and inference algorithms?
- Does the proposed next sequence prediction objective (Eq 3) correspond to a valid NELBO?
- Can the authors show the importance of contiguous block decoding via ablation?
- Can the authors show the effectiveness of their approach relative to multi-token prediction [2,3]?
- Can the authors detail the differences between the proposed parallel block diffusion training and the vectorized training from [1]?
- In addition, the choice of $\tau \in \[ 0.82, 0.98 \]$  in Table 1 seems heuristic; can the authors clarify how $\tau$ was picked?

[1] Arriola et al. Interpolating between autoregressive and diffusion language models. ICLR 2025

[2] Cai et al. Medusa: Simple llm inference acceleration framework with multiple decoding heads. ICML 2024

[3] Gloeckle et al. Better & Faster Large Language Models via Multi-token Prediction. arXiv 2024

---

> ### Author Response · Authors · 2025-11-21
>
> We sincerely thank the reviewer for valuable comments and provide our responses and clarifications below.
>
> **Is NSP corresponds to a Valid NELBO**
>
> In continuous diffusion models, the NELBO combines KL and reconstruction terms over a multi-step noise process. In discrete token spaces, this formulation cannot be applied directly, and prior work on discrete/masked diffusion (e.g., MaskGIT [1], MD4 [2]) has shown that the variational objective can be rewritten as a weighted sum of cross-entropy losses. MD4 notes that the continuous-time objective becomes "a simple weighted integral of cross-entropy losses."
>
> Following this standard simplification, discrete DLMs adopt a **cross-entropy** based single-step denoising objective, which SDLM also uses. Our procedure consists of a full-mask forward step producing the discrete analogue of $x_T$, and a single-step block-wise recovery. The left-to-right NSP factorization inside each block is simply our chosen parameterization of the denoiser, rather than a modification of the underlying diffusion objective.
>
> While SDLM does not implement multi-step continuous noise schedules, we believe it remains consistent with the established discrete diffusion framework, where the NELBO naturally reduces to cross-entropy–based denoising.
>
> [1] Chang, Huiwen, et al. "Maskgit: Masked generative image transformer." CVPR 2022.
>
> [2] Shi, Jiaxin, et al. "Simplified and generalized masked diffusion for discrete data." NIPS 2024.
>
> **Pseudocode for training and inference**
>
> We now include pseudocode for SDLM training and inference in Appendix B. Regarding the reviewer’s two questions:
> 1. Random block positions. We uniformly sample a prefix positions, mask the next block to form $z_T$, and predict $z_0$ in one step. The block-wise attention mask ensures correct visibility for any pos. This randomization also helps the model learn to decode from arbitrary positions.
> 2. Random block insertion. Since sequence lengths vary, inserting one masked block avoids padding overhead and exposes the model to more diverse contexts, improving robustness to different decoding start points.
>
> **Contiguous block decoding**
>
> While full-sequence DLMs can model richer non-causal interactions, SDLM adopts contiguous block decoding to remain aligned with ALMs and compatible with KV-cache efficiency. Within each block, we still use bidirectional attention, allowing non-causal refinement. Ablations replacing this with causal attention reduce decodeing speed, confirming that SDLM benefits from diffusion-style within-block interactions even under contiguous decoding.
>
> **The effectiveness of SDLM relative to MTP**
>
> MTPs generate draft tokens and then verify them. Medusa and Eagel-3 use multi-head predictors, while SDLM produces a full block via diffusion denoising, enabling parallel generation with bidirectional attention. Our added comparisons in Appendix D show that SDLM matches the speedup of MTPs and speculative decoding while providing comparable performance. It also remains competitive with Eagel-3, and we expect further gains as DLM infrastructure improves.
>
> **The diffenence between SDLM and BL3-DM**
>
> Our goal is to convert an AR LM into a diffusion LM with minimal cost. This leads to several differences from BD3-LM:
> 1. We keep causal attention for already-decoded tokens, leveraging the strong prefix modeling ability of ALMs.
> 2. We train denoising only for fully masked blocks (i.e., BD3-LM at t=1), since inference uses a single denoising step with longest-prefix decoding and thus only encounters fully masked blocks.
> 3. We randomly sample the start position of the masked block during training, allowing the model to flexibly determine decoding lengths and handle arbitrary start positions at inference.
>
> **Overstated contribution Clarification**
>
> Our intention was only to note that current block diffusion LMs, such as BD3-LM, are demonstrated only at small scale (~110M) and require higher training cost and specialized noise schedules. Our contribution is a practical path: keep pre-training autoregressive and apply NSP/SDLM only at SFT, enabling block-style diffusion decoding through lightweight finetuning of a large AR checkpoint. We will revise the wording to avoid overstating the claim.
>
> **Presentation/clarity**
>
> We thank the reviewer for pointing out the presentation and clarity issues. The unclear definitions are revised and marked in light blue in the revised PDF.
>
> **Choice of $\tau$**
>
> In our experiments, $\tau$ is a tunable parameter rather than a fixed value. As shown in Fig. 3, we sweep $\tau$ to balance speed and performance. Table 1 highlights two representative choices:
> * 0.98 (near-SFT quality with ~2× speedup)
> * 0.82 (higher speed with a modest drop)
> allowing users to select the trade-off that best suits their needs.

---

### Author Response · Authors · 2025-12-03

Dear Area Chair,

We are deeply grateful for the time and energy you have dedicated to this process, particularly given the unique challenges and heavy workload of the current ICLR cycle.

To help streamline your decision, we summarize how we addressed the reviewers' three main post-rebuttal concerns:

**Is SDLM still a DLM**

Yes. SDLM directly follows the standard masked diffusion formulation for discrete data: (i) mask a block of tokens in the forward process, and (ii) denoise by predicting the original tokens from mask tokens in the reverse process.
Recent discrete diffusion work shows the diffusion NELBO reduces to a weighted sum/mixture of MLM cross-entropy losses, and SDLM’s objective matches this cross-entropy denoising form even without explicit timesteps—making SDLM a blockwise, single-step masked diffusion language model aligned with modern discrete diffusion models.

**How does SDLM compare to other speed-up methods (e.g., MTP)?**

We added extensive experiments (Appendix D) comparing SDLM with (a) MTP/speculative decoding methods (e.g., Medusa, EAGLE-3), (b) larger block sizes at inference, and (c) vLLM baselines.

Overall, SDLM achieves comparable or stronger throughput while offering simple parallel block-wise generation via diffusion denoising with self-speculative decoding.

Importantly, SDLM generalizes to larger block size (d) at inference than training, even without retraining, remaining stable while improving throughput—so pJp5’s concern about larger D is not an issue in our experiments.

**How does SDLM differ from BD-3LM, and does it help?**

Our goal is to retrofit an autoregressive LM into a diffusion-style LM at lower cost, so we: (a) keep causal attention to decoded prefixes, (b) train only on fully masked blocks (consistent with single-step inference), and (c) randomize block start positions for flexibility. We also
ran BD-3LM-style objectives in early exploration and found SDLM with our next sequence prediction scheme (i.e., longest prefix decoding)  better preserves AR capabilities and yields stronger overall results under similar training budgets.

In addition, we implemented the requested manuscript revisions from reviewers Thez and pJp5. For reviewer TpGK, we provide a separate, detailed response covering training cost and performance analyses of math tasks.

Best regards,

The Authors

---

### Meta-Review · Area_Chair_jevm · 2026-01-07

**Summary:**

The paper proposes SDLM to accelerate LLM decoding via block-wise parallel generation. The main concerns were (i) whether SDLM is meaningfully a diffusion model given single-step full masking, (ii) missing MTP/speculative baselines, (iii) unclear novelty vs block diffusion / BD-3LM.

**Reviewer Concerns:**

1. The authors argue SDLM is still a masked discrete diffusion model because recent theory reduces the diffusion NELBO to a (mixture of) cross-entropy MLM losses, and SDLM matches this in single-step full-mask form.
2. They added Appendix D comparisons to MTP/spec decoding (Medusa, EAGLE-3), plus vLLM TPS baselines, claiming comparable throughput. Their main novelty over speculative decoding seems to be extending the block size at inference time.
3. They distinguish from BD-3LM via: keeping causal prefix attention, training only on fully masked blocks, randomizing block starts, and reporting early experiments suggesting SDLM better preserves AR behavior under similar budgets.

I think the new baselines weaken the contributions of this paper. SDLM performs just as well as MTP/speculative decoding (Eagle3) and even seems slightly slower at 8 and 16 block sizes (SuP). The only benefit offered is inference time block extension, which is not a big enough contribution to warrant acceptance: one could easily train an MTP model with a larger block size.

**Reviewer Scores:**

While the reviewers would have acknowledged the resolved issues, I don't believe they would increase scores based on the results of speculative decoding.

---

### Decision · Program_Chairs · 2026-01-26

Reject